# Combined Prospective Seroconversion and PCR Data of Selected Cohorts Indicate a High Rate of Subclinical SARS-CoV-2 Infections—an Open Observational Study in Lower Saxony, Germany

Rebecca Jonczyk,[a] Nils Stanislawski,[b] Lisa K. Seiler,[a] Holger Blume,[b] Stefanie Heiden,[c] Henning Lucas,[c] Samir Sarikouch,[d] Philipp-Cornelius Pott,[e] Meike Stiesch,[e] Corinna Hauß,[f] Giulietta Saletti,[g] Mariana González-Hernández,[g] Franziska Karola Kaiser,[g] Guus Rimmelzwaan,[g] Albert Osterhaus,[g] Cornelia Blume[a]

[a]Institute of Technical Chemistry, Leibniz University Hannover, Hannover, Germany
[b]Institute of Microelectronic Systems, Leibniz University Hannover, Hannover, Germany
[c]Institute of Innovation Research, Technology Management & Entrepreneurship, Leibniz University Hannover, Hannover, Germany
[d]Department of Cardiac, Thoracic, Transplantation, and Vascular Surgery, Hannover Medical School, Hannover, Germany
[e]Clinic for Dental Prosthetics and Biomedical Materials Science, Hannover Medical School, Hannover, Germany
[f]MVZ Labor Limbach Hannover GbR, Lehrte, Germany
[g]Research Center for Emerging Infections and Zoonoses, University of Veterinary Medicine Foundation, Hannover, Germany

Rebecca Jonczyk and Nils Stanislawski contributed equally to this publication. Author order was determined by the engagement of each author.

**ABSTRACT** Despite lockdown measures, intense symptom-based PCR, and antigen testing, the SARS-CoV-2 pandemic spread further. In this open observational study conducted in Lower Saxony, Germany, voluntary SARS-CoV-2 PCR tests were performed from April 2020 until June 2021, supported by serum antibody testing to prove whether PCR testing in subjects with none or few symptoms of COVID-19 is a suitable tool to manage the pandemic. In different mobile stations, 4,817 subjects from three different working fields participated in the PCR testing. Serum antibody screening using the SARS-CoV-2 ViraChip IgG (Viramed, Germany) and the Elecsys Anti-SARS-CoV-2 assay (Roche, Germany) was performed alongside virus neutralization testing. Subjects were questioned regarding comorbidities and COVID-19 symptoms. Fifty-one subjects with acute SARS-CoV-2 infection were detected of which 31 subjects did not show any symptoms possibly characteristic for COVID-19. An additional 37 subjects reported a previous SARS-CoV-2 infection (total prevalence 1.82%). Seroconversion was discovered in 58 subjects with known SARS-CoV-2 infection and in 58 subjects that never had a positive PCR test. The latter had a significantly lower Charlson Comorbidity Index, and one third of them were asymptomatic. In 50% of all seroconverted subjects, neutralizing serum antibodies (NAbs) were detectable in parallel to N/S1 ($n = 16$) or N/S1/S2 antigen specific antibodies ($n = 40$) against SARS-CoV-2. NAb titers decreased within 100 days after PCR-confirmed SARS-CoV-2 acute infection by at least 2.5-fold. A relatively high rate of subclinical SARS-CoV-2 infections may contribute to the spread of SARS-CoV-2, suggesting that in addition to other intervention strategies, systematic screening of asymptomatic persons by PCR testing may significantly enable better pandemic control.

**IMPORTANCE** Within this open observational study, repeated PCR ($n > 4,700$) and antibody screening ($n > 1,600$) tests were offered in three different working fields. The study identified 51 subjects with acute SARS-CoV-2 infection and 37 subjects reported to have had a positive PCR test taken externally. Thirty-one of the 51 subjects did not display any symptoms prior to testing. In addition, 58 subjects without PCR-confirmed SARS-CoV-2 infection were identified by seroconversion. Subjects, that had undergone SARS-CoV-2 infection without having noticed, more often had a

Address correspondence to Cornelia Blume, blume@iftc.uni-hannover.de.

The authors declare no conflict of interest.

low grade of immunization with no NAbs, but may have relevantly contributed to the spread of the pandemic. Based on these results, we suggest that both regular PCR and rapid test screening of symptomatic and asymptomatic individuals, specifically within groups or workplaces identifiable as having close quarter contact, thus increased infection transference risk, is necessary to better assess and therefore reduce the spread of a pandemic virus.

**KEYWORDS** COVID-19, subclinical cases, PCR, SARS-CoV-2 antibody screening, neutralizing antibodies, antibody screening tests, working groups

Preventive non-pharmaceutical intervention strategies such as implementing quarantine measures cause societal and economic disruption, largely due to the negative consequences of social distancing and strict isolation of high-risk groups (1). For example, in Italy, strict isolation of segments of the population has led to an extensive reduction in coronavirus disease 2019 (COVID-19) positive subjects and those with severe COVID-19 progression (2), but has had substantial negative economic and social ramifications. The overall effect of different intervention strategies during the pandemic in Germany cannot yet be fully assessed, but future pandemic containment concepts should probably rely on more targeted information on the spread of the pandemic in different populations. The Robert Koch Institute (RKI) initially recommended rapid SARS-CoV-2 PCR testing only in case of exposure to a SARS-CoV-2 infected person or in the presence of COVID-19 symptoms (3). Despite that, unintended testing in wider population groups was heavily discussed (4).

PCR tests from nasopharyngeal or throat swabs were predominantly used to detect acute SARS-CoV-2 infections. According to a recent meta-analysis, these PCR tests are above 70% sensitive (sensitivity: 73.3%, 95% CI = 68.1%, 78.0% specificity) and above 95% specific (5). With a low estimated prevalence of the disease in the general population of less than 3% (6), the positive predictive value is limited with relatively low test-sensitivity. Specific antibody detection can provide a more accurate picture of previously experienced infections. Antibodies can be detected at the earliest one to 2 weeks after infection (6). Infection with SARS-CoV-2 induces antibody responses against several SARS-CoV-2 proteins, including the spike protein (S protein) and the nucleocapsid (N) protein, which both become detectable in the median second week after symptom onset (7). It is important to study the kinetics of the antibody response against SARS-CoV-2 and the order in which antigen specific antibodies rise. Several studies have shown that SARS-CoV-2 infections elicit a robust antibody response, which consists mainly of antibodies of the IgG and IgM classes, of which those directed to the S or N proteins of the virus are most relevant for population studies (8). Seroconversion of antibodies against SARS-CoV-2 usually occurs 7 to 14 days after symptom onset (8, 9). In most cases, virus specific IgM antibodies are detectable first, followed by IgGs (10), although in some cases this order may vary (11). NAbs targeting the S protein which interfere with viral entry (12, 13) are usually detectable by the end of the second week after symptom onset (14, 15), albeit not in all subjects (16).

This study presents the results of a serological screening into the prevalence of SARS-CoV-2 specific antibodies indicating past prevalence of a SARS-CoV-2 infection. In addition, an assessment of seroconversion in these subjects was established to estimate more precisely whether the measures taken in Germany since March 2020 were effective enough to stop the spread of SARS-CoV-2. Seropositive subjects were identified and in addition, in these subjects NAbs were investigated, because they reflect the level of protection from SARS-CoV-2 infection or disease. Subjects belonged to three study populations ("education/ culture," "company," "nursing homes" in Lower Saxony, Germany). The study addresses aspects of mobile deployment, rapid set-up for local and timely pandemic diagnostics.

## RESULTS

**Study participants.** In total, 4,817 subjects underwent PCR testing, of these 4,701 subjects (m = 2,589; f = 2,112) participated in PCR testing (Table 1, Fig. 1) and

Microbiology Spectrum

**TABLE 1** Overview about the numbers of subjects who participated in the SARS-CoV-2 PCR testing and in the antibody screening of the study

| Diagnostic measure | Education/culture | | | Company | | | Nursing homes | | |
|---|---|---|---|---|---|---|---|---|---|
| | m | f | Total | m | f | Total | m | f | Total |
| PCR | 641 | 884 | 1,525 | 1,850 | 932 | 2,782 | 98 | 296 | 394 |
| PCR and antibody screening | 367 | 533 | 900 | 381 | 315 | 696 | 42 | 129 | 171 |
| Only antibody screening | 41 | 57 | 98 | 0 | 0 | 0 | 6 | 12 | 18 |

additional 116 study participants (m = 47, f = 69) reported to have had PCR tests at external testing centers. Testing was performed for the main part of participants between April 2020 and November 2020 at two container based test centers, while mobile teams continued testing in some subgroups (e.g., educational institutions) until June 2021. Subjects accepted the offer for PCR testing with varying frequencies. The number of PCR tests performed within the study ranged widely between one and 46 tests per subject, median four. About 27% of the subjects underwent PCR testing only once during the several months of the study period.

Figure 2 summarizes the age and gender distribution of all subjects and shows that the majority of participants in all groups was female, but the numbers of either PCR positive or seroconverted subjects was not significantly different in one gender-group ($P \geq 0.05$ according to chi-square test for independence). There were significant age differences between the groups: mean age was significantly higher in nursing homes versus the two other groups, and in the education/culture group versus the company ($P < 0.0001$ according to Mann-Whitney-U test).

Altogether, 1,883 subjects (m = 837; f = 1,046) participated in antibody screening tests (Table 1, Fig. 1). Among those, 1,767 subjects also participated in the PCR tests of the study, with a median value of nine PCR tests taken per subject, whereas 116 subjects had undergone PCR testing in other settings. Antibody screening tests started in September 2020 and were performed until June 2021. Participating subjects had not been vaccinated against SARS-CoV-2.

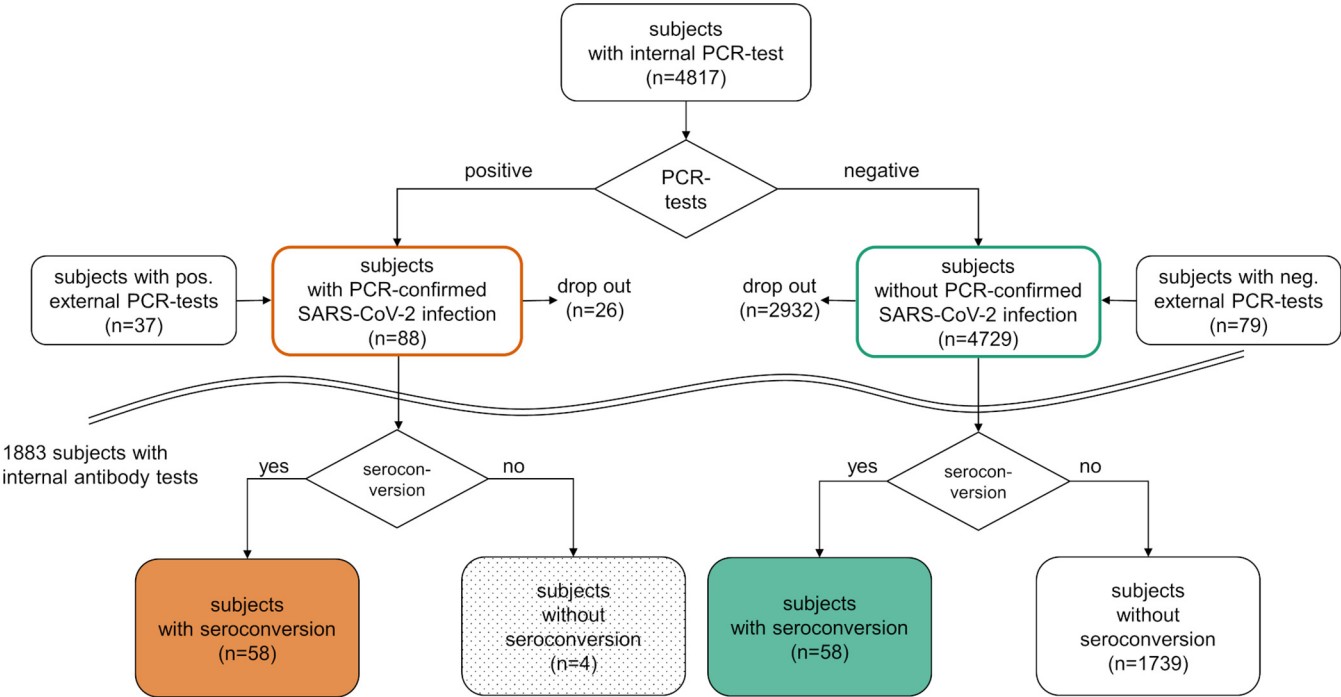

**FIG 1** Schematic displaying all subjects in the study undergoing PCR tests with positive and negative results and all subjects with antibody tests with positive or negative results. Twenty-six subjects with a positive PCR test did not participate in antibody screening ("dropout").

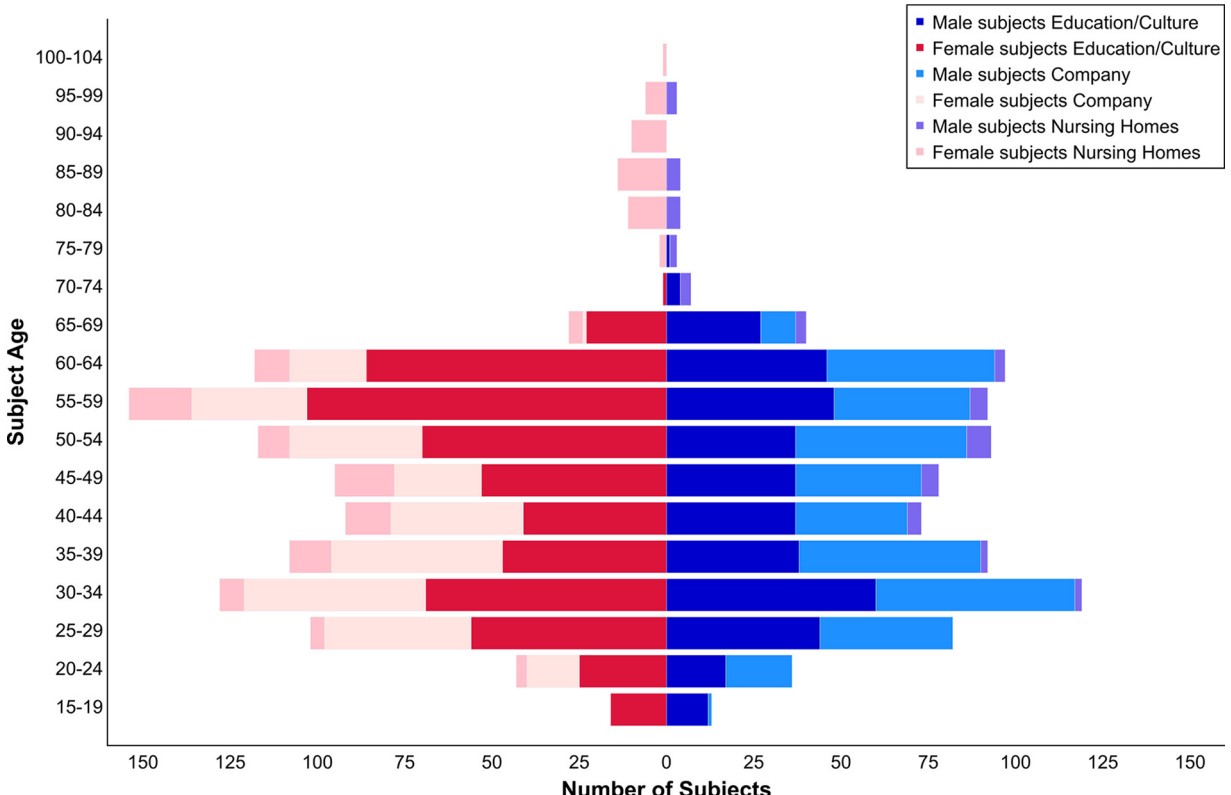

**FIG 2** Schematic depicting age and gender distribution among all participating groups of the study. Three groups were defined: (i) education/culture with employees out of universities, schools, city administration of Hannover, theater; (ii) company with employees from a biotechnological company in Goettingen; and (iii) nursing homes with residents and employees of two nursing homes in Hannover.

**Acute SARS-CoV-2 infections identified by PCR.** PCR testing and data acquisition displayed a prevalence of 1.97% within the "Education/Culture" group between April 2020 and June 2021, a prevalence of 0.43% within the "company" group between April 2020 and November 2020 and 11.68% within the "nursing homes" group between April 2020 and January 2021. In the group depicted as "nursing homes," one big cluster of infections was documented in a single nursing home by PCR testing within several weeks in December 2020. The prevalence for this outbreak within this specific time frame was 13.75%.

Subjects reported acute symptoms according to the questionnaires (supplement Table S1, questionnaire I) at each test. Thirty-one of the 51 subjects tested positive for an acute SARS-CoV-2 infection at one of the established test centers did not report any possibly characteristic COVID-19 symptoms in the past 14 days. These 31 asymptomatic subjects displayed a significantly ($P = 0.002$ according to Mann-Whitney-U test) lower virus load determined by higher Ct-values as opposed to symptomatic subjects (Fig. 3). For asymptomatic subjects the mean Ct-value was 31.84 (SD = 4.35), for symptomatic subjects 27.4 (SD = 4.53).

**SARS-CoV-2 seroconversion identified by antibody screening and timely appearance of antibody patterns after a positive PCR-proof of acute infection.** Serum antibody testing revealed a total number of 116 seroconverted subjects, including 58 subjects after a positive PCR test as well as 58 subjects without a previous positive PCR test (Fig. 4). In addition, four subjects failed to show seroconversion after being tested positive by PCR test. One of these subjects had a relatively low virus load (Ct-value of 36) while there was no information on the virus load of the three other subjects available.

In subjects with PCR-confirmed SARS-CoV-2 infections, the timely appearance of certain antibody patterns after the first positive PCR test (from now on referred to as

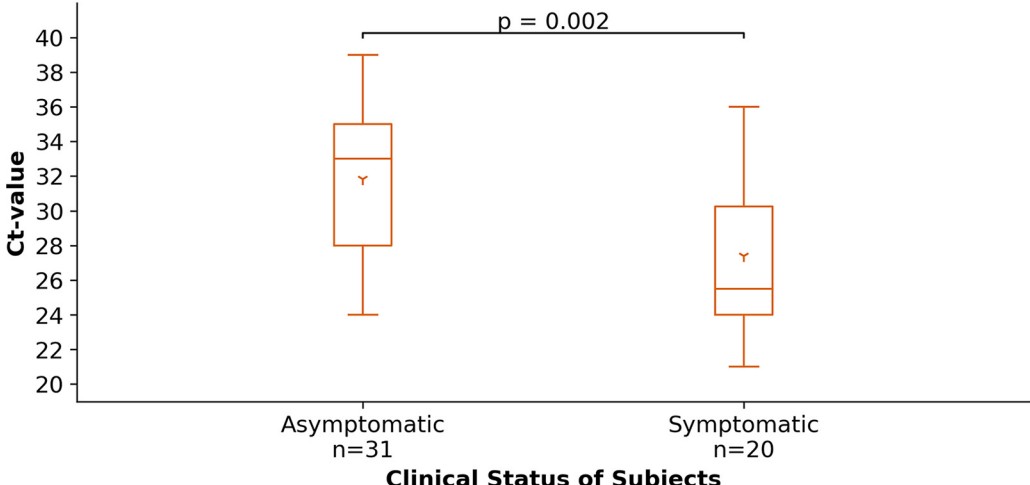

**FIG 3** Correlation between clinical status of subjects at the time point of positive PCR test and the Ct-value as corresponding factor of the virus load. All subjects had been asked whether they had experienced any of the following possibly characteristic COVID-19 related symptoms: cough/snuff, fever/shivering, sore throat, headache, limb pain, fatigue, shortness of breath, diarrhea, and smell/taste loss (Table S1).

time point 0 or "T0") could be stated (Fig. 5). Six serum samples were taken relatively early (up to day 34) after T0 and all tested negative in the respective assays. N-specific antibodies without S-specific antibodies in parallel were detected as early as 13 days after T0. S1-specific antibodies without N or S2 specific antibodies in parallel were recorded between 74 and 133 days after T0. A combination of N/S1 targeting antibodies appeared between day 21 and 327 after T0. Antibodies targeting the N/S1/S2 pattern of antigens were recorded within days 7 to 400 after T0. The predominant part of all subjects who have seroconverted with previous positive PCR detection of a SARS-CoV-2 infection showed either N/S1 specific antibodies or antibodies targeting the N/S1/S2 virus proteins.

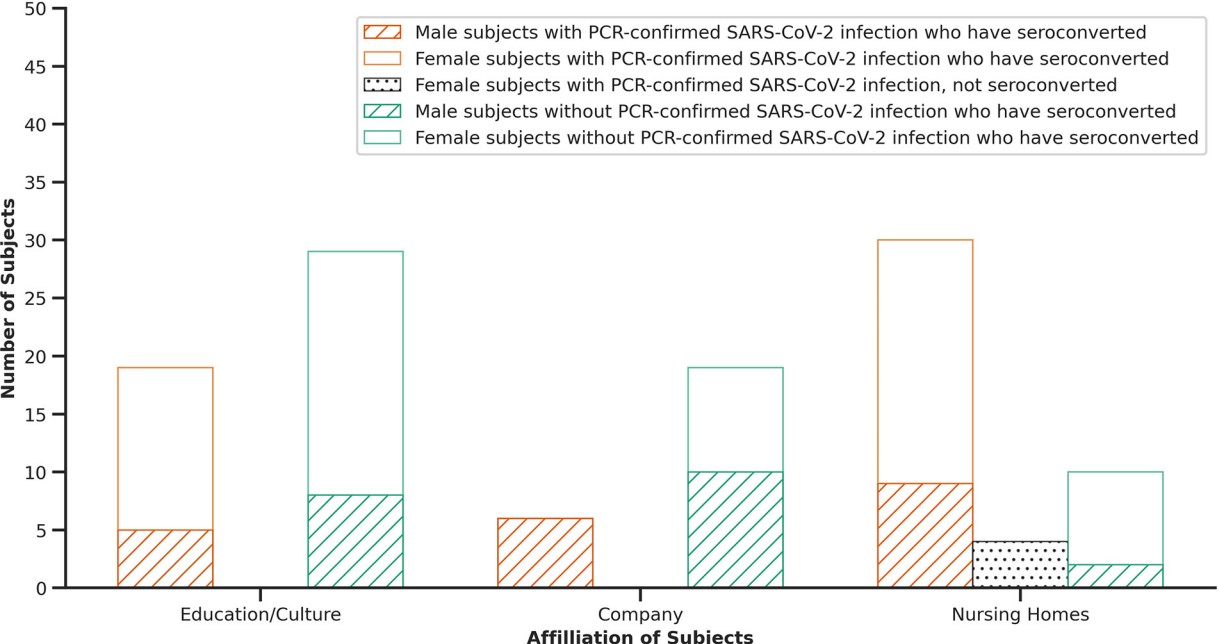

**FIG 4** Numbers of seroconverted male and female subjects with PCR-confirmed SARS-CoV-2 infection (orange, female = white, male = striped), recognized by PCR test, in each group (education/culture; company; nursing homes) and numbers of seroconverted male and female subjects despite no previous positive SARS-CoV-2 PCR test (female = white, male = striped). Not seroconverted female subjects with PCR-confirmed SARS-CoV-2 infection are displayed as dotted column.

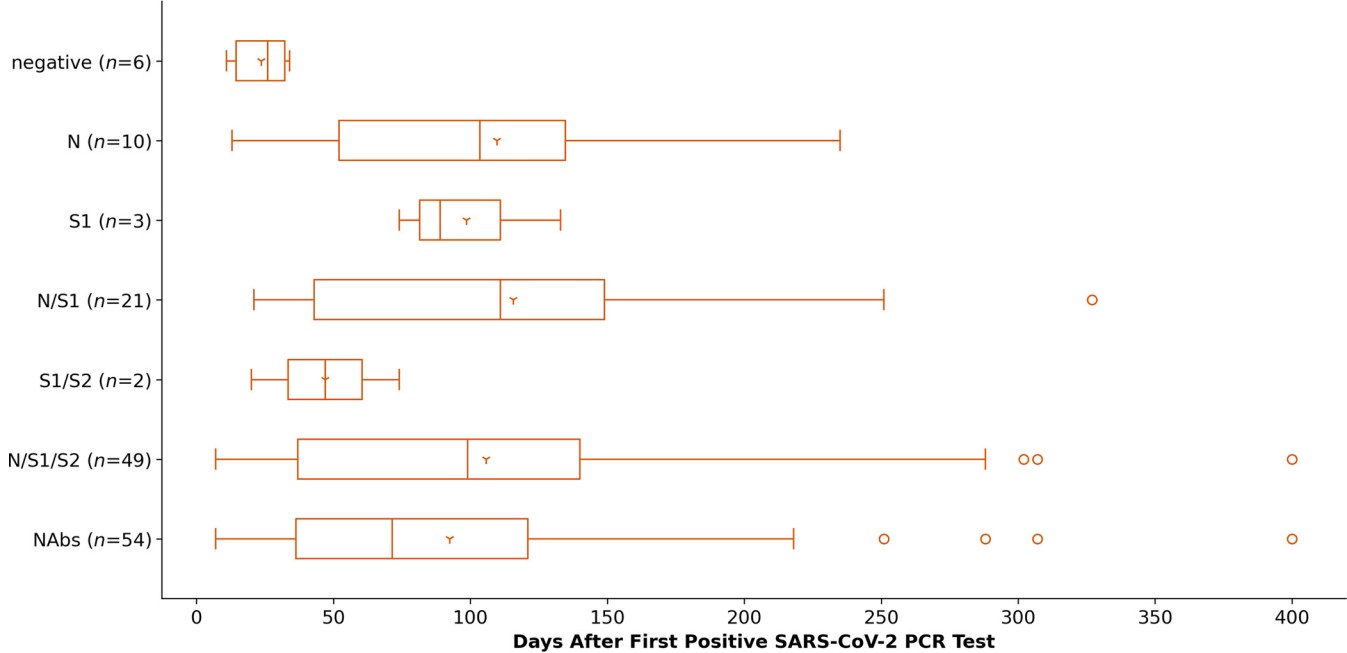

**FIG 5** Time related occurrence of antigen-specific antibodies (N, S1, S2) as well as neutralizing antibodies (NAbs) after first positive PCR test for SARS-CoV-2 in days.

**Correlation of symptoms and antibody patterns.** Of the 58 subjects who had a previously PCR-confirmed SARS-CoV-2 infection and participated in antibody screening, four experienced an apparently asymptomatic course of infection. A 58-year-old male subject who had undergone a symptomatic SARS-CoV-2 infection with interstitial pneumonia (proven by computer tomography), fever, snuff, cough, and postinfection air shortage for several months, showed N/S1/S2 antigen-specific antibodies and NAbs 400 days after the first positive SARS-CoV-2 PCR test taken. After PCR-confirmed SARS-CoV-2 infection, four subjects did not develop SARS-CoV-2 specific antibodies as shown in Elecsys and ViraChip and VN tests, performed within 34 days after T0. Ct-values obtained from at least one of these subjects displayed a low virus load at T0. Three out of four of these subjects were more than 85 years old and had a Charlson Comorbidity Index of 7 or 8.

Antibody screening identified 58 subjects without a previous PCR-confirmed test for SARS-CoV-2 infection who had seroconverted. Of these subjects, 5.5% stated pre-existing chronic conditions (e.g., lung disease, diabetes, and heart disease) in comparison with 44% in the group with a confirmed positive PCR test. Altogether, the group without PCR-confirmed SARS-CoV-2 infection had a significantly ($P < 0.05$ according to Mann-Whitney-U test) lower comorbidity profile according to the Charlson Comorbidity Index, as indicated by the data acquired by the questionnaires used (Fig. 6).

**Possible humoral immunity in seroconverted SARS-CoV-2 subjects.** VN tests were performed on antibody positive serum samples of both subjects with or without PCR-confirmed SARS-CoV-2 infection. Fig. 7 shows NAb titers grouped according to their antibody pattern obtained by the antibody assays and their antigen-specificities. NAbs both emerged more often and with higher titers in samples from subjects who displayed antibodies against S1/S2/N-antigens. Subjects who developed antibodies against N/S1 antigens always displayed NAbs. Most samples from subjects with only antibodies against S2, S1, N, N/S2 respectively, or no antibodies against any of these antigens failed to show NAbs. Subjects without PCR-confirmed SARS-CoV-2 infection showed fewer or none NAbs compared with subjects with PCR-confirmed SARS-CoV-2 infection.

One single serum sample from a subject with PCR-confirmed SARS-CoV-2 infection was tested negative 22 days later in both the Elecsys and ViraChip but did display

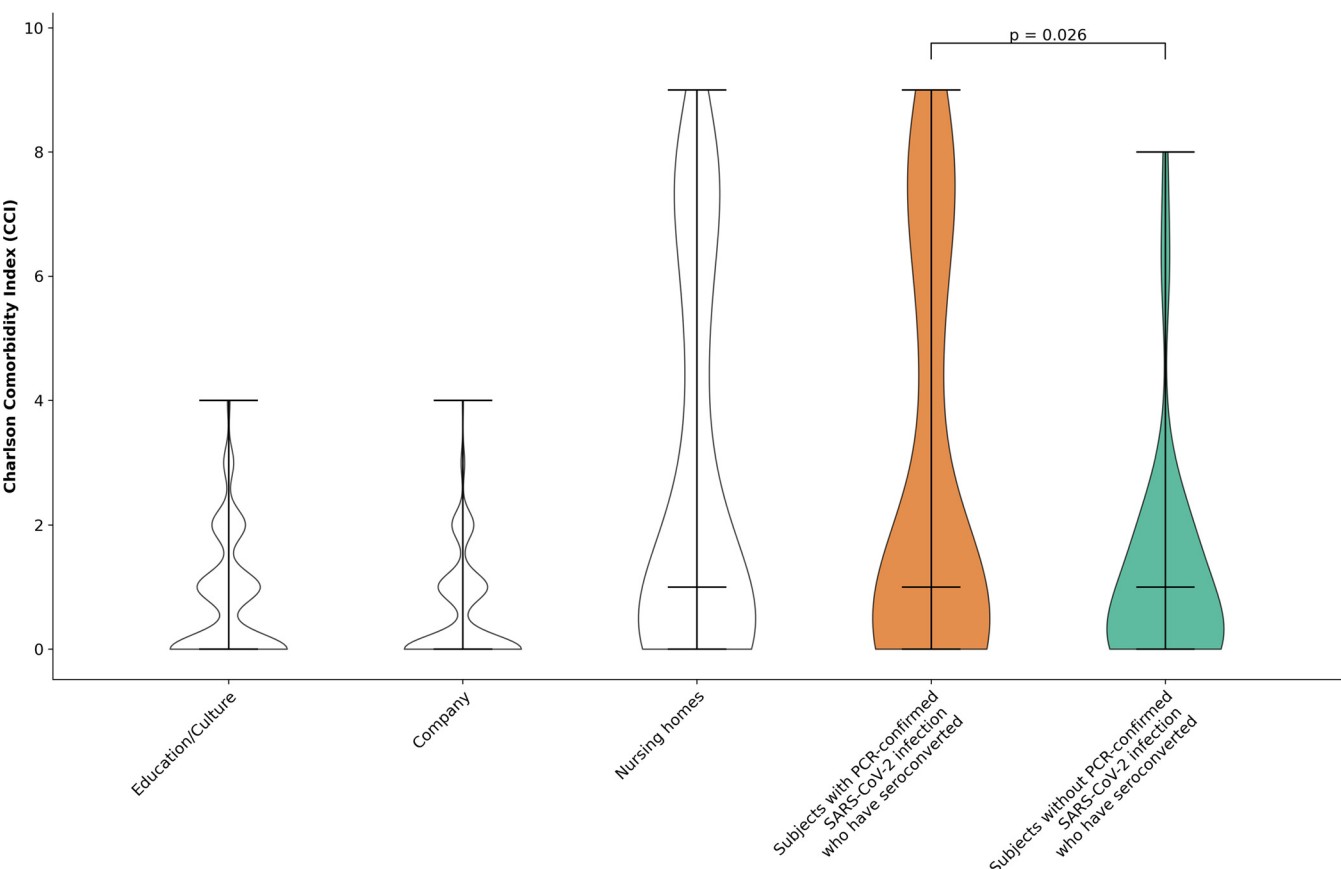

**FIG 6** Charlson Comorbidity Index for subjects within the groups "educational/culture," "company," and "nursing homes" as well as the subgroups of subjects with and without PCR-confirmed SARS-CoV-2 infection who have seroconverted. Statistical significance was calculated using Mann-Whitney-U test.

NAbs. This subject however displayed a pattern of N/S1/S2 antigen specific antibodies 105 days later. In one subject with PCR-confirmed SARS-CoV-2 infection and S1/S2 serum antibodies, no NAbs could be demonstrated. These results were found as late as 365 and 379 days after T0.

Figure 8 displays seropositive subjects identified with the respective serological assays with and without PCR-confirmed SARS-CoV-2 infection identifying an apparent subgroup of NAb-non-responders. Subjects with only N antigen specific antibodies often did not display NAbs and neither did subjects without PCR-confirmed SARS-CoV-2 infection with solely S1 or solely S2 specific antibodies. Samples from subjects with PCR-confirmed SARS-CoV-2 infection and no antibodies found in the immunoassay did not display NAbs. Furthermore, in the group of subjects who tested seropositive by any of the immunoassays, but without prior positive PCR test, VN test results were more often negative.

In selected antibody-positive serum samples from subjects with PCR-confirmed SARS-CoV-2 infection, NAbs were detected as early as day 7 and as late as day 400 (Fig. 5). Fig. 9 displays the decreasing titers of NAbs in seroconverted subjects with known SARS-CoV-2 infection, 10 of these had a second VN test (68 samples were analyzed by VN tests) and their NAb course is depicted using dashed lines. As early as within the first 100 days after T0, NAb-titers decreased by at least 2.5-fold in these subjects.

In line with this, antigen specific antibody profiles and detection of NAbs differed distinctly from the profiles of subjects with PCR-confirmed SARS-CoV-2 infection (Fig. 8): 30 of these 58 subjects presented either N/S1/S2 or N/S1 antigen specific antibodies in parallel to NAbs. Another 10 of these 58 subjects only had N specific antibodies, six subjects had S1 specific antibodies, six subjects had S2 specific antibodies, five

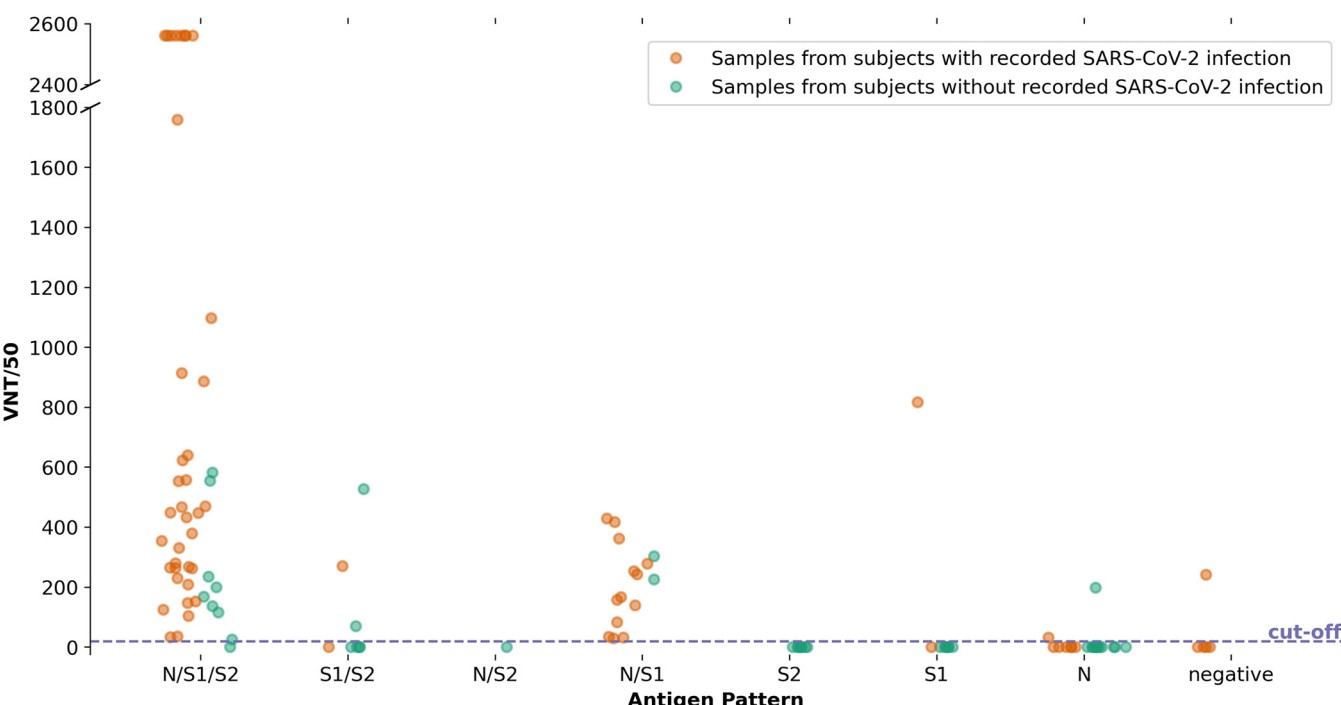

**FIG 7** Virus neutralization test results of each serum sample based on antigen-specific antibody patterns obtained by Elecsys and ViraChip.

subjects had S1/S2 specific antibodies, and one subject had N/S2 specific antibodies. However, out of these 28 subjects only three displayed NAbs.

About half of the subjects without PCR-confirmed SARS-CoV-2 infection reported— based on the questions (supplement Table S1, questionnaires I, II, and III) asked at each

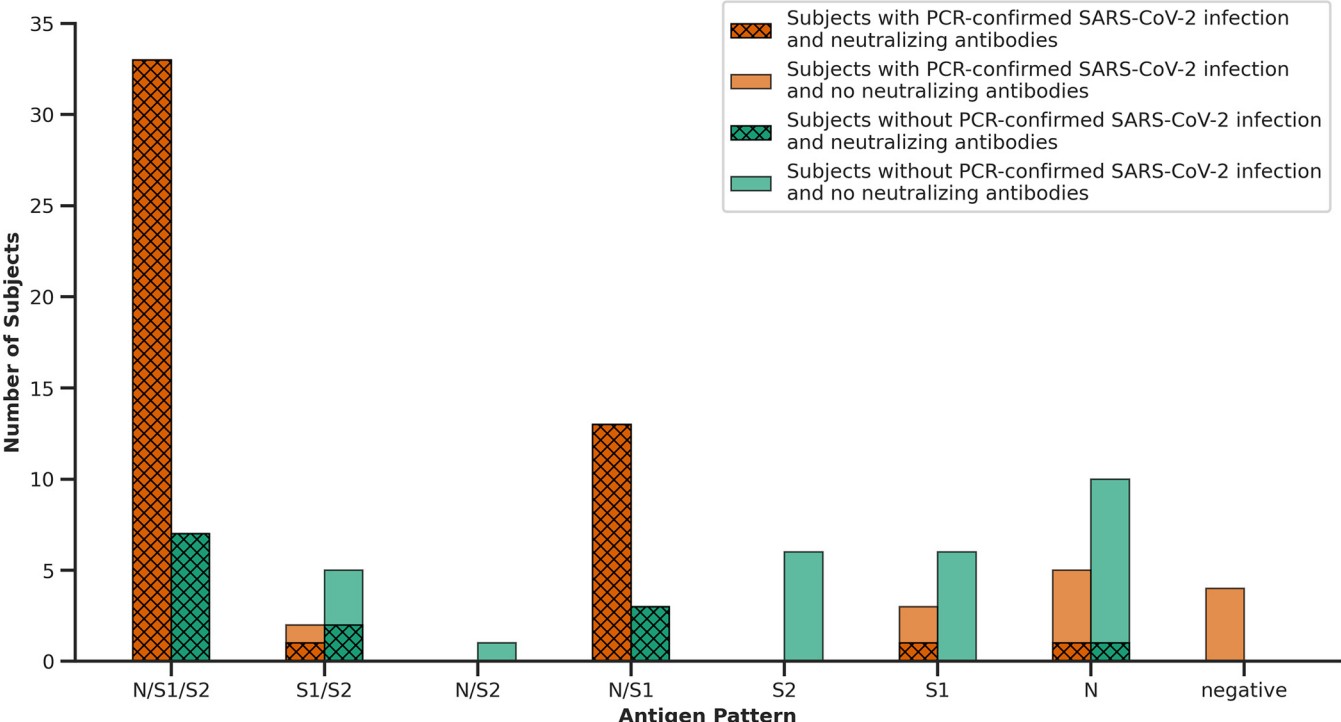

**FIG 8** Detection of antigen specific antibodies (N, S1, S2) in all PCR-positive subjects (orange) and in all subjects without a recorded positive SARS-CoV-2 PCR test (green). A positive virus neutralization-test result is depicted by a hatched overlay.

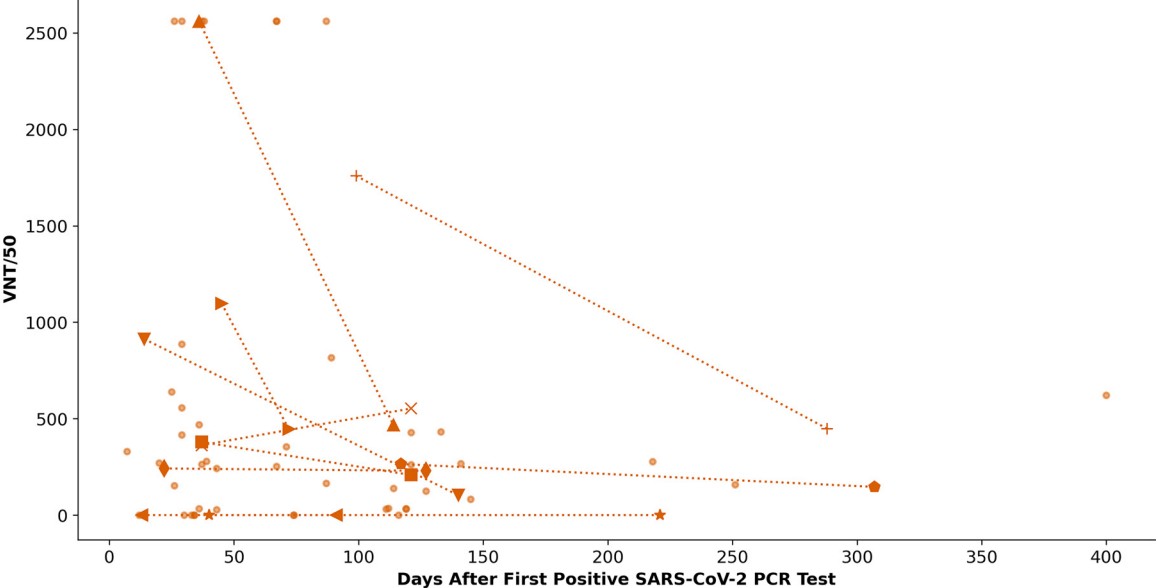

**FIG 9** Virus neutralization titers (VNT) of neutralizing antibodies over time were expressed as the reciprocal of the dilution that gave a 50% reduction of stained cells. Samples from subjects with more than one sample are highlighted and connected with dotted lines.

testing for acute and passed COVID-19 symptoms—to have had a period of varying symptoms since the beginning of the COVID-19 pandemic that could be attributed to a SARS-CoV-2 infection (Fig. 10). Among the symptoms reported by subjects with only one symptom, individually occurring symptoms of fever/shivering, cough/snuff, sore throat, taste/smell loss, or headache were reported.

## DISCUSSION

Whether or not SARS-CoV-2 screening in asymptomatic subjects is justified, has been widely discussed. It was suggested that people with asymptomatic SARS-CoV-2 infection in close and continuous contact with infectious index cases were less infectious than symptomatic cases due to a more rapid seroconversion (17). In contrast, a publication of September 2020 reviewing the symptomatic state of several cohorts worldwide estimated that especially these asymptomatic persons account for approximately 40% to 45% of SARS-CoV-2 infections (18).

The study presented here was performed to further clarify this divergence and to prove whether not-for-cause PCR testing in a pandemic situation is useful and necessary—regardless of the material and personnel resources that must be afforded. In this study, there was a varying and inconsistent attendance of subjects in PCR testing, for reasons that may be manifold (e.g., disregard of symptoms or asymptomatic infection, home office, no test recommended by the general practitioner even despite of symptoms, unsuitable test procedure).

In the present study, 4,817 subjects from the public life of Hannover and Goettingen ("education"/"company") as well as from two nursing and retirement homes ("nursing homes") in Hannover were selected through their voluntary participation. These study participants underwent repeated PCR screening irrespective of their clinical status and antibody screening to identify a previous SARS-CoV-2 infection. The study offered repeated PCR and antibody screening tests over the course of April 2020 until June 2021. About 60% of subjects tested positive for an acute SARS-CoV-2 infection by PCR at one of the test stations provided within this study's framework did not report any common COVID-19 symptoms and would not have been identified without the test offer. More than 50% of all study participants who had tested positive by serum antibody screening, had apparently undergone unidentified, largely subclinical SARS-CoV-2

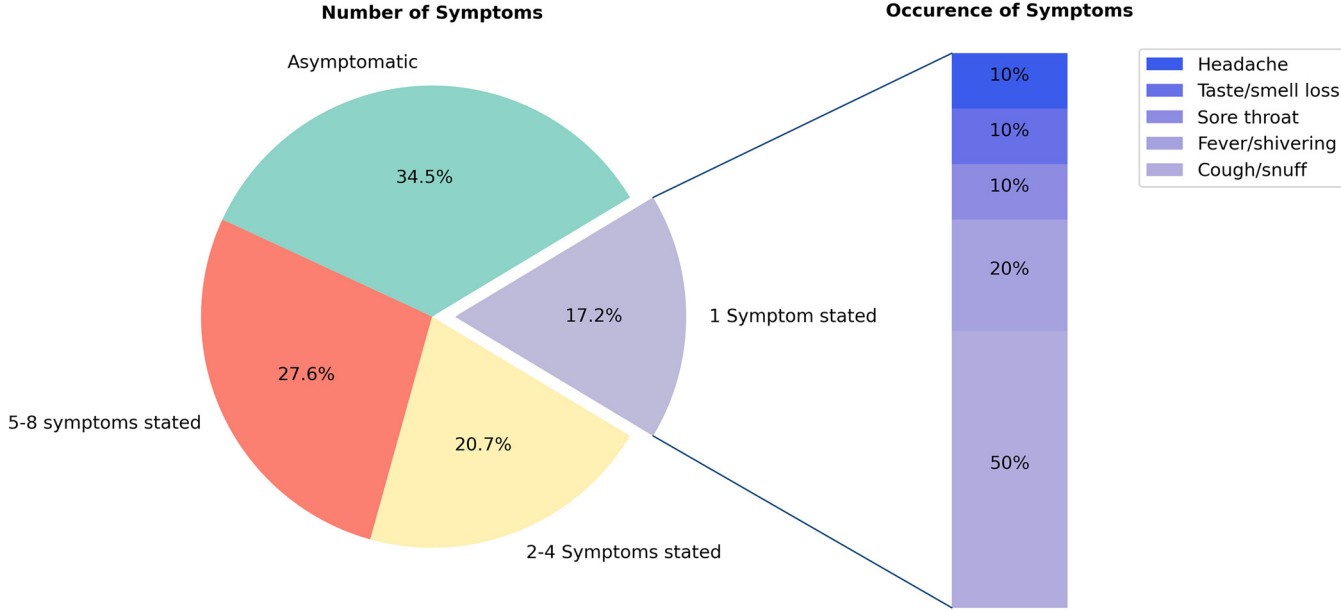

**FIG 10** Percentage of subjects with either none, 1, 2 to 4 or 5 to 8 COVID-19 typical symptoms among all seroconverted subjects without PCR-confirmed SARS-CoV-2 infection. All subjects had been asked whether they had experienced any period since the onset of the corona crisis with at least one of the following symptoms, which the subjects themselves related to a possible SARS-CoV-2 infection: cough/snuff, fever/shivering, sore throat, headache, limb pain, fatigue, shortness of breath, diarrhea, and smell/taste loss (Table S1).

infections. That is a significant finding as it proves the estimated two to three times higher infection rate in the general population.

The PCR-stated prevalence of SARS-CoV-2 infections from the educational and cultural institutions and that of the "company" group were less than 3% and therefore in accordance with the prevalence of SARS-CoV-2 infection in Lower Saxony in January 2021 (1.74%; 139.023 cases [19] per ∼ 8 million inhabitants in Lower Saxony [20]). In contrast, the prevalence in the nursing homes tested was higher (mean of 9.35%, reaching a maximum of 13.75% during an outbreak in a single nursing home), which is characteristic for this group with a high risk for virus spreading due to its high number of comorbid individuals and decreased possibility for social distancing (21). Regarding the results from the antibody testing, the PCR-confirmed prevalence of 1.97% had to be corrected to 3.87% for the group "education/culture," from 0.43% to 1.11% for the group "company" and from 9.35% to 14.21% for the group "nursing homes." This is only explicable by the fact the subjects withdrew from regular testing which was not recommended by experts, and this was reflected by a low individual test frequency.

Based on these results, we suggest, that asymptomatic persons should be included in regular PCR screening to limit the spread of the COVID-19 pandemic. Meanwhile, numerous rapid antigen tests based on pharyngeal or nasal swabs have been developed as a cheap and fast alternative, that are less sensitive, but still useful. According to a recently published paper, a rapid antigen test reached a test sensitivity of up to 89.6% when applied to samples with viral loads typically seen in infectious patients (22). Rapid antigen tests may be of limited benefit, as demonstrated by a false-negative rate of up to 45% of antigen testing when performed by laypersons (23). In addition, also PCR tests can be false-positive (24). Here, strict PCR testing must be coupled with strict precautionary measures and possibly repeated PCR tests in ambiguous cases with regard to sample contamination, in order to keep the number of false-positive results as low as possible, because a false-positive result can have considerable economic and social ramifications. Summarizing, future pandemic intervention scenarios should weigh to what extent intense and combined PCR and serum antibody testing may be an important additional tool to limit the spread of the infection in the workplace. This procedure might support the consideration of an "inverse quarantine" postulated by others (1).

The timely appearance of antigen specific antibodies in subjects with PCR-confirmed SARS-CoV-2 infection in this study revealed that N specific antibodies occur earlier from day 13 on (Elecsys, probably IgM positivity), whereas S specific antibodies without N specific antibodies were detected as late as from day 52 after PCR test on. This is in line with results of others (25).

At present, no general statements can be made about the role of T cell mediated immunity against SARS-CoV-2 upon the natural course of infection, but a closer look might be valuable because experience with other coronavirus diseases (SARS and MERS) suggests that this immunity could last as long as 3 years (26–29). SARS-CoV-2 NAbs are suggested to be associated with convalescence, and they have been detected at least as long as 12 months after disease (30). In the study presented here, these antibodies were detectable as long as 400 days after PCR-confirmed SARS-CoV-2 infection in a 58-year-old male subject.

The correlation of antibody patterns in terms of antigen specificity and the presence of NAbs has also previously been discussed. Beadoin et al. described, that effective NAbs are present only within a very early time interval after a SARS-CoV-2 infection. They observed that the levels of receptor-binding-domain (RBD)-specific IgG and IgA slightly decreased between 6 and 10 weeks after the onset of COVID-19 symptoms and that RBD-specific IgM levels decreased much more abruptly (31). In the data presented, NAbs were present even until day 400 after a positive PCR result. Predominantly, their occurrence correlated with an antibody pattern consisting of either N/S1/S2 or at least N/S1. In eight subjects with only N antibodies, the VN tests were only positive in three cases supporting the idea that S1 antibodies correlate with the virus neutralizing activity (32). All of the seroconverted subjects with PCR-confirmed SARS-CoV-2 infection displayed VN antibodies, whereas only 13 out of 58 subjects without PCR-confirmed SARS-CoV-2 infection but seroconversion tested with immunoassays displayed NAbs, which confirms a lower immunization grade in subjects with only a few symptoms. Reasons for that could be, that higher NAb titers appear to be associated with more severe disease also involving response by extranodal B lymphocytes (33) or that the infection in these subclinical cases occurred too far in the past.

Among the seropositive subjects without PCR-confirmed SARS-CoV-2 infection, 30 remembered none or only a single COVID-19 symptom whereas only four subjects with PCR-confirmed SARS-CoV-2 infection were asymptomatic. Besides a lower-level complex of COVID-19 typical symptoms, the 58 subjects who had seroconverted despite no previous positive PCR test showed a significantly lower comorbidity profile than the group with PCR-proven SARS-CoV-2 infection. The presence of a high comorbidity may be a confounding factor for symptomatic SARS-CoV-2 infection (34–36). This might support the assumption that they were not aware of COVID-19 and therefore did not ask (or were not recommended to have) specific PCR testing. As much as 20 of these subjects never had experienced any symptoms since the beginning of the pandemic.

**Summary.** This open observational study—performed among subjects of three different working fields with repeated PCR ($n > 4,700$) and antibody screening ($n > 1,600$) tests—revealed 51 subjects with acute SARS-CoV-2 infection, 31 subjects testing PCR positive were asymptomatic for the last 14 days, and 58 subjects with not consciously experienced SARS-CoV-2 infection were only discovered by their serological status. The number of subjects with a history of SARS-CoV-2 infection identified only by antibody and not by PCR screening was comparable to the number of PCR positively tested subjects. This led to an upwardly corrected prevalence rate in the populations tested.

Based on these study results, we suggest that both regular PCR and antigen test screening of both symptomatic and asymptomatic individuals, specifically within groups or workplaces identifiable as having close quarter contact and thus have an increased infection transference risk, is necessary to better assess and therefore reduce the spread of a pandemic virus.

## MATERIALS AND METHODS

**Selection of participants and ethical approval.** This is an open observational epidemiological study to assess medical and virological parameters in different groups of subjects. Testing within this study did neither interfere with ongoing intervention strategies nor interact with any parallel intervention study participation. For these studies, the "Aerztekammer Niedersachsen" issued an ethical approval in August 2020 (No. Bo/30/2010; Bo/31/2010; Bo/31/2010). Study subjects belonged to three different groups located in the southern part of Lower Saxony, Germany: The first group, defined as "education/culture," included employees from universities, theater, schools, administration in and close to Hannover, Germany. The second group, defined as "company," consisted of employees of a biotechnological company in Goettingen, Germany. The third group included both employees and residents of retirement and nursing homes in and close to Hannover (designated as "nursing homes").

The respective employers or heads of each institution or company recommended the subjects to participate. The following RKI guidelines were used to invite and select subjects: being over 50 years of age, having comorbidities such as diabetes, obesity, cardiac, pulmonary, circulatory or immunocompromising diseases, or taking immunosuppressants. Although the PCR testing was offered to asymptomatic subjects with regard to infection risk at the test center, some individuals conferred acute symptoms possibly characteristic for COVID-19 at the day of testing. Furthermore, individuals in non-medical occupational groups were preferentially considered to be particularly exposed if they had many contacts with people during the corona crisis due to their occupational activities (working in service areas, frequent customer contact or abundant contact with colleagues). In nursing and retirement homes, the administration of the institution gave a recommendation for the persons to be tested according to their risk of exposure. Persons, who received the offer to be tested, could anonymously refuse to participate without giving a reason. Furthermore, questionnaires were used to register from the beginning of the COVID-19 pandemic whether COVID-19 symptoms or risks regarding SARS-CoV-2 infection had been present (supplement Table S1). Before sample collection, subjects were informed about the PCR and antibody testing and asked for their study consent. Subjects, which were identified as being seroconverted due to a SARS-CoV-2 infection with one of the two antibody test systems were asked for additional blood samples to test for NAbs within 14 days.

**PCR screening tests.** For PCR testing, a pharyngeal smear was performed by swabbing the region between the anterior and posterior palatal arches on the subject. Most throat swabs were performed at a mobile screening facility by trained medical personnel after informed consent of the subject. RNA purification was performed using a viral RNA minikit from Qiagen (QIAamp Viral RNA minikit; MD, USA) or Thermo Fisher (MagMAX Viral and Pathogen Nucleic Acid Isolation Kits; Waltham, USA) according to manufacturer's instruction. PCR was performed according to the guidelines valid at the time of execution. Therefore, primers amplifying the nucleocapsid region of SARS-CoV-2 (N1 and N2) or the envelope region (E) and RNA-dependent RNA polymerase (RdRp) of SARS-CoV-2 described by Corman et al. (37) and U.S. CDC (2019-nCoV CDC EUA kit), respectively, were used. SARS-CoV-2 specific PCR was conducted in triplicates and classified as positive if two of three results were positive. Sufficiency of RNA was proven by a marker (human RNase P), samples with not sufficient material were repeated. Results were quantified as described in Pfaffl et al.(38). Cycle threshold values (Ct-values) for classification of a positive PCR test were corresponding to a virus titer of 250 virus particles.

**Antibody screening tests.** Blood samples were taken from the cubital vein, kept vertically for 30 min and stored at 4°C for less than 24 h before centrifugation. Serum was obtained by centrifugation of blood samples at 4,000 rpm for 10 min at room temperature (RT). Samples were either analyzed within 2 or 3 days after collection or after storage in cryotubes (1 mL) at −80°C (short time storage until analysis at 4°C).

For serum antibody detection, two different qualitative and complementary detection methods were used: the electro-chemiluminescence immunoassay Elecsys Anti-SARS-CoV-2 (Elecsys; Roche Diagnostics, Mannheim, Germany) and the microarray-based immunoassay SARS-CoV-2 ViraChip IgG Test Kit (ViraChip; Viramed Biotech AG, Planegg, Germany). Elecsys was used to detect IgM and IgG antibodies against the N virus antigen. The second test was performed utilizing the ViraChip test kit detecting IgG antibodies against S1, S2, and N virus antigens, which was evaluated with defined serum samples provided by Limbach AG, Luthe, Lower Saxony, as well as with serum samples collected before 2019 by the DRK-blood donor service Lower Saxony (data not shown). According to these tests, N antigen specific antibodies were screened with two different approaches (IgM/IgG versus only IgG). Seropositivity was declared when at least one of these antibody types was present.

**(i) Electro-chemiluminescence immunoassay elecsys anti-SARS-CoV-2.** This immunoassay was performed at cobas e411 rack using manufacturer's materials and reagents and according to manufacturer's instructions directly after serum preparation via centrifugation in serum gel tubes. For small sample volumes, Hitachi sample cups were used. After analyzing, samples were divided into aliquots and stored. Results were divided in negative (<0.9 cutoff index [COI]), borderline (0.9 to 1.1 COI) and positive (>1.1 COI). According to the manufacturer¨s recommendation, borderline ($n = 28$) as well as positive test results were categorized as positive.

**(ii) Microarray-based immunoassay SARS-CoV-2 ViraChip IgG.** Before test performance, serum samples in cryotubes were processed as follows: Samples were diluted by pipette robot (rLINE, Sartorius Stedim Biotech, Goettingen, Germany) equipped with 1-channel pipette head (1 to 10 $\mu$L) in a 96-well PCR plate (Sarstedt AG & Co.KG, Nümbrecht, Germany), using safety space low retention sterile filter tips (1 to 10 $\mu$L, Sartorius Stedim Biotech) and sample buffer included in the test kit. SARS-CoV-2 ViraChip IgG Pos Control and SARS-CoV-2 ViraChip IgG and IgM Neg Control were analyzed as positive and negative controls. Positive and negative controls were diluted according to manufacturer's instructions 1:16

(10 $\mu$L control serum + 150 $\mu$L sample buffer). In 30 samples of SARS-CoV-2 tested persons (12 positive/ 18 negative) from MVZ Labor Dr. Limbach, antibody-positive results were first-line borderline-weak. These were twice repeated and the dilution of samples was optimized to ensure a reproducible test result of samples, so these could be unambiguously classified as positive or negative. Samples of known test results were used for this purpose in a concentration row (2, 4, 6, 8, 10 $\mu$L serum + 150 $\mu$L sample buffer). The dilution of samples was finally adjusted to 6 $\mu$l of serum mixed with 150 $\mu$l sample buffer. Since it was observed, that freshly unfrozen samples showed lower signal intensities than samples stored at 4°C overnight before measurement, all further measurements were performed within 2 to 4 days after thawing or immediately after sample collection.

All further proceedings were according to manufacturer's instructions. All washing and incubation steps were performed at RT and 750 rpm (VWR, Darmstadt, Germany). To add and remove the washing solutions, a Hydroflex Washer (Tecan, Crailsheim, Germany) was used. Test readout was performed with ViraChip Scanner and analyzed with ViraChip Software (Viramed Biotech AG, Planegg, Germany). Results were divided into negative (<69 ViraChip units), borderline (69-99 ViraChip units) and positive ($\geq$100 ViraChip units for antibodies against one antigen S1, S2, or N) results. According to the manufacturer's recommendation both borderline as well as positive test results were categorized as positive.

**(iii) SARS-CoV-2 virus neutralization test.** NAbs were determined in 80% of seropositive subjects as described before (39, 40). In brief, serum samples were inactivated at 56°C for 30 min diluted first 1:20 in cell culture medium followed by 2-fold serial dilutions and mixed with 200 TCID50 of SARS-CoV-2 strain Wuhan-Hu-1. Serum-virus mix was incubated at 37°C for 1 h, then added to Vero cell monolayers and incubated for 8 h at 37°C, 5% $CO_2$. Afterwards, cells were fixed with 4% paraformaldehyde followed by 80% methanol. For detection of SARS-CoV-2, plates were blocked for 30 min using 1% BSA, followed by 1 h of incubation with a 1:1,000 dilution of rabbit polyclonal anti-SARS-CoV-2 nucleocapsid (SinoBiological). Then, cells were washed with PBS-0.05% Tween 20, a 1:1,000 dilution of anti-rabbit-IgG-Alexa Flour 488 (Invitrogen) was added and cells were further incubated at room temperature for 30 min. Finally, cells were washed twice and fluorescent cells were counted using the C.T.L. S6 Ultimate-V Analyzer and data was analyzed using CTL ImmunoSpot software. Virus neutralization titers were expressed as the reciprocal of the dilution that gave a 50% reduction of stained cells (VNT/50). Titers of 20 and lower were considered negative.

**Data acquisition and statistical evaluation.** Before swab or blood samples were taken, subjects were asked to answer questions (supplement Table S1, questionnaires I and II) regarding their comorbidities, acute symptoms, and the likelihood of having had SARS-CoV-2 infection in the past. In addition, positive PCR tests from other testing centers, as well as the subject's vaccination status were noted. All subject and sample related information were acquired using a proprietary software and stored in a secure and encrypted database.

Charlson Comorbidity Index was calculated (41) based upon the acquired subject's information (supplement Table S1, questionnaire I). Statistical evaluation was performed with Python. Data was tested for normality using D'Agostino's K-squared test and chi-square test for independence was used to test whether distributions differ. Mann-Whitney-U test was used for analysis of not normally distributed data.

**Quality assurance and lab safety standards.** Sample collection and treatment as well as PCR and antibody analyses were performed according to the guidelines of German Medical Association for quality assurance of examinations in laboratory medicine (42) and an External Quality Assurance Services (EQAS) program was successfully accomplished. In case of a borderline result during antibody testing, the analysis was repeated within a time frame of 16 h. Samples with a persisting borderline result were classified as positive ($\sim$12%) according to manufacturer's recommendations.

**Data availability.** De-identified subject data sets will be available upon written request to the corresponding author following publication.

## SUPPLEMENTAL MATERIAL

Supplemental material is available online only.
**SUPPLEMENTAL FILE 1**, PDF file, 0.1 MB.

## ACKNOWLEDGMENTS

The study was financed by state funds from the Ministry of Economics of Lower Saxony. The sponsor did not exert any influence or make any recommendation as to which groups of people should be tested. The offer of testing was requested by various institutions or groups of persons themselves.

G.S. and G.R. received funding from the Alexander von Humboldt Foundation in the framework of the Alexander von Humboldt Professorship. The work at the Research Center for Emerging Infections and Zooneses was financially supported by the Ministry of Research and Culture of Lower Saxony (COFONI and MWK). F.K.K. was funded by the DFG (German Research Foundation)—398066876/GRK 2485/1 (VIPER).

We further thank several PhD students at the Institutes for Technical Chemistry and Microelectronic Systems (Leibniz University Hannover, Germany) for intense work on both PCR and antibody testing as well as technical and organizational support in this

project. Furthermore, we thank the medical and dental students from the Medical School Hannover (MHH) and from the university in Goettingen for their support registering subjects and taking swabs. We thank Nils Hoppe and his team for excellent support of the study's ethical vote and procedures with regard to data protection (Centre of Ethics and Law in the Life Sciences, Leibniz University, Hannover, Germany).

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
