## [Reviewer comments · Microbiology Spectrum]

Microbiology Spectrum

Combined prospective seroconversion and PCR data of selected cohorts indicate a high rate of subclinical SARS-CoV-2 infections - an open observational study in Lower Saxony, Germany

Rebecca Jonczyk, Nils Stanislawski, Lisa Seiler, Holger Blume, Stefanie Heiden, Henning Lucas, Samir Sarikouch, Philipp Pott, Meike Stiesch, Corinna Hauss, Giuletta Saletti, Mariana Gonzalez-Hernandez, Franziska Kaiser, Guus Rimmelzwaan, Albert Osterhaus, and Cornelia Blume

Corresponding Author(s): Cornelia Blume, Leibniz University Hannover

Review Timeline:

Submission Date:	September 10, 2021
Editorial Decision:	October 25, 2021
Revision Received:	December 7, 2021
Editorial Decision:	December 20, 2021
Revision Received:	December 24, 2021
Accepted:	January 10, 2022

Editor: Kileen Shier

Reviewer(s): The reviewers have opted to remain anonymous.

Transaction Report:

DOI: <https://doi.org/10.1128/spectrum.01512-21>

October 25, 2021

Prof. Cornelia Anneliese Blume
Leibniz University Hannover
Institute for Technical Chemistry
Callinstrasse 3-5
Hannover, Lower Saxony 30167
Germany

Re: Spectrum01512-21 (Combined prospective seroconversion and PCR data of selected cohorts indicate a high rate of subclinical SARS-CoV-2 infections - an open observational study in Lower Saxony, Germany)

Dear Prof. Cornelia Anneliese Blume:

Thank you for submitting your manuscript to Microbiology Spectrum. When submitting the revised version of your paper, please provide (1) point-by-point responses to the issues raised by the reviewers as file type "Response to Reviewers," not in your cover letter, and (2) a PDF file that indicates the changes from the original submission (by highlighting or underlining the changes) as file type "Marked Up Manuscript - For Review Only". Please use this link to submit your revised manuscript - we strongly recommend that you submit your paper within the next 60 days or reach out to me. Detailed information on submitting your revised paper are below.

Link Not Available

Sincerely,

Kileen Shier

Journals Department
Reviewer comments:

Reviewer #1 (Comments for the Author):

Minor clarifications:

1. pg 13, under Acute SARS-CoV 2 infections identified by PCR; why are the PCR testing and data acquisition time frames different for each subgroups? Could you also clarify why the manuscript went on to state that the prevalence in the 'nursing home' subgroup was 13.75% at the end of the paragraph, whereas it was 11.68% in the sentences prior. Did you mean that for the month of December 2021, prevalence was up to 13.75%?
2. pg 13, under SARS-CoV-2 seroconversion identified by antibody screening; "In addition, few subjects failed to show seroconversion after being tested positive by PCR test." Could be helpful to put a number to it.
3. pg 14, "In one subject with PCR-confirmed SARS-CoV-2 infection and S1/S2 serum antibodies, no NABs could be demonstrated. These results were found as late as 365 and 379 days after a SARS-CoV-2 positive PCR test." - Interesting observation, but was there a particular reason for testing NABs on 365 and 379 day specifically, after a positive PCR test?
4. One of the limitations of the study that is not addressed is the retrospective nature of interviewing seropositive and PCR

negative patients; you would be relying on patients remembering their likelihood of having SARS-CoV 2 infection in the past, coupled with a moderately small sample size of 58, it could skew your interpretation of results.

Main feedback;

The manuscript has indeed highlighted the value of PCR testing in asymptomatic SARS-CoV 2 patients, however, with regards to the suggestion of regular PCR test as a screening tool, more considerations have to be made. As mentioned earlier in the manuscript, PCR testing may have a high specificity of 95% and over, however, if carried out in a large population size with a low prevalence of infection, it would translate to a significant number of false positives, resulting in unnecessary economic and social ramifications for these individuals (Surkova, E., Nikolayevskyy, V., Drobniewski, F., 2020. False-positive COVID-19 results: hidden problems and costs. *The Lancet Respiratory Medicine* 8, 1167-1168.. doi:10.1016/s2213-2600(20)30453-7). Much thought has to be given to balance the pros and cons of such a screening programme.

Well written manuscript and this observational study will be of keen interest to readers.

Reviewer #2 (Comments for the Author):

Jonczyk et al report an observational study about occurrence of sub-clinical infections, with negative RT-PCR. In addition, the authors described the pattern of antibodies in relation also to the functional response (nAb). The paper is informative, and investigate an hot-topic for COVID-19 like sub-clinical infections.

However, my main suggestion is to revise, shorten and reorganise the paper in order to simplify the information about the antibodies patterns and go to the key messages that the authors want to underline. I suggest the use of tables.

In addition, some other comments:

- Please clarify, the individuals tested are all asymptomatic at the time of diagnosis?
- The individuals present co-morbidities, can this represent a confounding factors in the immune response detected?
- Do you have information of viral load (Ct values) in seroconverted individuals and in who did not show seroconversion?
- How many subjects over 58 (few) failed to show seroconversion? This is an important controversial aspect of regular screening.
- Have you repeated the serological testing on longitudinal samples for those individuals who did not seroconverted?
- Please, describe better the proportion of seroconverted subjects (i.e. % over the total seroconversion, over the total of the tested individuals,...). If we count over the total tested subjects, I do not see high rate of subclinical infections. Please, clarify your conclusions.
- Have you tested possible cross-reactivity for other coronaviruses in seroconverted PCR-negative subjects?

Staff Comments:

Preparing Revision Guidelines

Please return the manuscript within 60 days; if you cannot complete the modification within this time period, please contact me. If you do not wish to modify the manuscript and prefer to submit it to another journal, please notify me of your decision immediately so that the manuscript may be formally withdrawn from consideration by Microbiology Spectrum.

If your manuscript is accepted for publication, you will be contacted separately about payment when the proofs are issued;

please follow the instructions in that e-mail. Arrangements for payment must be made before your article is published. For a complete list of **Publication Fees**, including supplemental material costs, please visit our website.

Microbiology Spectrum

Paper Submission Jonczyk et al.

“Combined prospective seroconversion and PCR data of selected cohorts indicate a high rate of subclinical SARS-CoV-2 infections”

Point-to-point Response to Reviewers

Comments from reviewers are formatted bold, our responses are colored blue. All page numbers stated within this document refer to the “Marked Up Manuscript – For Review Only” document provided alongside this response to reviewers.

Reviewer #1 (Comments for the Author):

Minor clarifications:

1. pg 13, under Acute SARS-CoV 2 infections identified by PCR; why are the PCR testing and data acquisition time frames different for each subgroups?

The study was based upon voluntary participation. It started in April 2020 as a free offer. Starting with two PCR test centers located in containers, one in Hannover, one in Goettingen, the main part of participants came in first from educational and cultural institutions, mainly from our university, and from the company in Goettingen. In parallel, mobile teams performed PCR testing in nursing homes in and next to Hannover. Another nursing home with mobile teams came in December 2020; single groups at schools were tested by mobile teams until June 2021.

A clarification of these circumstances was made on page 12: *“Testing was performed for the main part of participants between April 2020 and November 2020 at two container*

Prof. Dr. med. Cornelia Blume (apl)
Institute of Technical Chemistry
Callinstr. 5
D-30167 Hannover
hannover.de

E-mail: blume@iftc.uni-hannover.de
Phone: +49 (0)511 762-2963
Fax: +49 (0)511 762-3004
Website: <https://www.tci.uni->

based test centers, while mobile teams continued testing in some subgroups (e.g. educational institutions until June 2021)."

Could you also clarify why the manuscript went on to state that the prevalence in the 'nursing home' subgroup was 13.75% at the end of the paragraph, whereas it was 11.68% in the sentences prior. Did you mean that for the month of December 2021, prevalence was up to 13.75%?

The prevalence rate was given for all nursing home participants as a total and also for those who were part of the subgroup with the mentioned "infection outbreak". Here the rate of infections was higher.

This is now clarified on page 13: "*The prevalence for this outbreak within this specific time frame was 13.75%.*"

2. pg 13, under SARS-CoV-2 seroconversion identified by antibody screening; "In addition, few subjects failed to show seroconversion after being tested positive by PCR test." Could be helpful to put a number to it.

There were four subjects with no immune response, this is now mentioned on page 14 (we replaced "few" with "four") and page 15.

3. pg 14, "In one subject with PCR-confirmed SARS-CoV-2 infection and S1/S2 serum antibodies, no NAbs could be demonstrated. These results were found as late as 365 and 379 days after a SARS-CoV-2 positive PCR test." - Interesting observation, but was there a particular reason for testing Nabs on 365 and 379 day specifically, after a positive PCR test?

This individual appeared for antibody testing to find out, whether he/she was still showing antibodies after their SARS CoV-2 infection more than 300 days before. The second blood drawing was performed to test the presence of NAbs after a positive response from the test after 365 days for antibodies.

This information on a general procedure within the study is now added on page 7: *"Subjects, which were identified as being seroconverted due to a SARS-CoV-2 infection with one of the two antibody test systems were asked for additional blood samples to test for NAbs within 14 days."*

4. One of the limitations of the study that is not addressed is the retrospective nature of interviewing seropositive and PCR negative patients; you would be relying on patients remembering their likelihood of having SARS-CoV 2 infection in the past, coupled with a moderately small sample size of 58, it could skew your interpretation of results.

Each subject in the study was questioned for possible COVID-19 symptoms at each time point when he or she performed PCR testing. In addition, all subjects undergoing antibody testing were additionally asked for having ever experienced any symptoms typical for COVID-19, therefore the information was not purely retrospectively collected. See also page 11: we added *"acute"* symptoms to better describe the questions asked at each test: *"Before swab or blood samples were taken, subjects were asked to answer questions (supplement Table S1, questionnaires I and II) regarding their comorbidities, acute symptoms, and the likelihood of having had SARS-CoV-2 infection in the past."*

For additional clarification, we also added the information about the concrete use of questionnaires on page 17: *"About half of the subjects without PCR-confirmed SARS-CoV2 infection reported – based on the questions (supplement Table S1, questionnaire I, II and III) asked at each testing for acute and passed COVID-19 symptoms - to have had a period of varying symptoms since the beginning of the COVID-19 pandemic that could be attributed to a SARS-CoV-2 infection (Figure 10)."*

Main feedback

The manuscript has indeed highlighted the value of PCR testing in asymptomatic SARS-CoV 2 patients, however, with regards to the suggestion of regular PCR test as a screening tool, more considerations have to be made. As mentioned earlier in the manuscript, PCR testing may have a high specificity of 95% and over, however, if carried out in a large population size with a low prevalence of infection, it would translate to a significant number of false positives, resulting in unnecessary economic and social ramifications for these individuals (Surkova, E., Nikolayevskyy, V., Drobniowski, F., 2020. False-positive COVID-19 results: hidden problems and costs. *The Lancet Respiratory Medicine* 8, 1167-1168. doi:10.1016/s2213-2600(20)30453-7). Much thought has to be given to balance the pros and cons of such a screening programme.

We added this remark to our discussion and cited the mentioned article on page 20: "*In addition, also PCR tests can be false positive (30). Here, strict PCR testing must be coupled with strict precautionary measures and possibly repeated PCR tests in ambiguous cases with regard to sample contamination, in order to keep the number of false-positive results as low as possible, as a false-positive result can have considerable economic and social ramifications. Summarizing, future pandemic intervention scenarios should weigh to what extent intense and combined PCR and serum antibody testing may be an important additional tool to limit the spread of the infection in the workplace.*"

Well written manuscript and this observational study will be of keen interest to readers

Thank you for your kind feedback.

Reviewer #2 (Comments for the Author):

Jonczyk et al report an observational study about occurrence of sub-clinical infections, with negative RT-PCR. In addition, the authors described the pattern of antibodies in relation also to the functional response (nAb). The paper is informative, and investigate an hot-topic for COVID-19 like sub-clinical infections.

However, my main suggestion is to revise, shorten and reorganise the paper in order to simplify the information about the antibodies patterns and go to the key messages that the authors want to underline. I suggest the use of tables.

From page 14 on, we reorganized the paper using the following headlines

-SARS-CoV-2 seroconversion identified by antibody screening and timely appearance of antibody patterns after a positive PCR-proof of acute infection

-Correlation of symptoms and antibody patterns

-Possible humoral immunity in seroconverted SARS-CoV-2 subjects

Rearranged text passages are highlighted yellow, shortened sentences are crossed out.

We've also replaced Figure 1 with a more detailed version to clarify the study's procedures and results.

In addition, some other comments: - Please clarify, the individuals tested are all asymptomatic at the time of diagnosis?

Although the tests were offered primarily to asymptomatic subjects, all subjects were asked for acute COVID-19 related symptoms at each testing time point. Their responses are part of the results presented within our paper.

For a better understanding we added information on page 6 regarding the selection of subjects: "*Although the PCR testing was offered to asymptomatic subjects with regard to infection risk at the test center, some individuals conferred acute symptoms possibly characteristic for COVID-19 at the day of testing.*"

We've also added an earlier reference to the questionnaires on page 13: "*Subjects reported acute symptoms according to the questionnaires (supplement Table S1, questionnaire I) at each test. 31 of the 51 subjects tested positive for an acute SARS-CoV-2 infection at one of the established test centers did not report any possibly characteristic COVID-19 symptoms in the past 14 days. These 31 asymptomatic subjects displayed a significantly ($p = 0.002$ according to Mann-Whitney-U test) lower virus load determined by higher Ct-values as opposed to symptomatic subjects (Figure 3). For asymptomatic subjects the mean Ct-value was 31.84 (SD: 4.35), for symptomatic subjects 27.4 (SD: 4.53).*"

The individuals present comorbidities, can this represent a confounding factors in the immune response detected?

The comorbidities can certainly represent a confounding factor in the immune response detected within subjects. This is a matter of debate. We have found several articles dealing with this question.

Paper 1: Comorbidity and its impact on Patients with COVID-19_Sanyaolu: Patients with comorbidities should take all necessary precautions to avoid getting infected with SARS-CoV-2, as they usually have the worst prognosis.

Paper 2: Epidemiological characterization of symptomatic and asymptomatic COVID-19 cases and positivity in subsequent RT-PCR tests in the United Arab Emirates_Al-Rifai: A significantly higher proportion of symptomatic cases had at least one chronic comorbidity than that of asymptomatic cases.

Paper 3: Asymptomatic SARS-CoV-2-infection_Sah: We also found that cases with comorbidities had significantly lower asymptomaticity compared to cases with no underlying medical conditions.

We've added this remark and the papers mentioned above to our discussion on page 22: "The presence of a high comorbidity may be a confounding factor for symptomatic SARS-CoV-2 infection (40–42)."

Do you have information of viral load (Ct values) in seroconverted individuals and in who did not show seroconversion?

We have added a statement regarding the correlation between Ct-values and symptoms for both seroconverted and one of the non-seroconverted subjects in the result section (page 13). This statement is backed by an additional figure (Figure 3), showing the Ct-values on the y axis for each of the groups of asymptomatic and symptomatic subjects.

Of the four subjects who did not show seroconversion only one subject was tested PCR-positive within our own laboratory. The other three subjects reported a previous SARS-CoV-2 infection tested at external facilities, which did not provide the corresponding Ct-values.

How many subjects over 58 (few) failed to show seroconversion? This is an important controversial aspect of regular screening. - Have you repeated the serological testing on longitudinal samples for those individuals who did not seroconverted?

On pages 14 and 15, four subjects non-responding with an antibody upregulation are mentioned, these subjects were formerly PCR positive despite the presence of few symptoms. The antibody testing was performed until day 34 after T0 (time point of first positive PCR-test), afterwards these subjects could not be considered any more since they were vaccinated.

Please, describe better the proportion of seroconverted subjects (i.e. % over the total seroconversion, over the total of the tested individuals,..). If we count over the total tested subjects, I do not see high rate of subclinical infections. Please, clarify your conclusions.

Out of the 4817 PCR-tested subjects, 4729 subjects (98.17%) were tested negative and 88 subjects (1.83%) were tested PCR positive (see figure 1). Within the 4729 negatively PCR-tested subjects, 58 showed seroconversion. Therefore, of the PCR-tested subjects 1.83% (88 subjects) were PCR-positive and additionally 1.2% (58 subjects) PCR-negative subjects

were subclinical cases. This means, about 2/3 of the identified subjects with a history for SARS-CoV-2 infection were detected by antibody and not by PCR testing, which is a high number.

This clarification was added to the paper on page 22 (summary): "*The number of subjects with a history of SARS-CoV-2 infection identified only by antibody and not by PCR screening was comparable to the number of PCR positively tested subjects. This led to an upwardly corrected prevalence rate in the populations tested.*"

Of the 51 subjects tested positive for a SARS-CoV-2 infection, 31 reported no possibly characteristic COVID-19 symptoms within the last 14 days at the time of testing. This information is revealed in the newly added figure (Figure 3), displaying the Ct-values of PCR tests for symptomatic and asymptomatic subjects. We also count these 31 subjects as subclinical cases and added the following information to the paper to emphasize this important conclusion.

On page 13: "*31 of the 51 subjects tested positive for an acute SARS-CoV-2 infection at one of the established test centers did not report any possibly characteristic COVID-19 symptoms in the past 14 days.*"

On page 19: "*About 60% of subjects tested positive for an acute SARS-CoV-2 infection at one of the test stations provided by this study's framework did not report any common COVID-19 symptoms and would not have been identified without the test offer.*"

On page 22 we inserted the underlined part in the summary: "*This open observational study – performed among subjects of three different working fields with repeated PCR ($n > 4700$) and antibody screening ($n > 1600$) tests – revealed 51 subjects with acute SARS-CoV-2 infection, 31 subjects tested PCR positive were asymptomatic for the last 14 days, and 58 subjects with not confirmed acute SARS-CoV-2 infection only discovered by their serological status."*

Have you tested possible cross-reactivity for other coronaviruses in seroconverted PCR-negative subjects?

Prof. Dr. med. Cornelia Blume (apl)
Institute of Technical Chemistry
Callinstr. 5
D-30167 Hannover
hannover.de

E-mail: blume@iftc.uni-hannover.de
Phone: +49 (0)511 762-2963
Fax: +49 (0)511 762-3004
Website: <https://www.tci.uni->

We didn't conduct any test for possible cross-reactivity for other coronaviruses in seroconverted PCR-negative subjects. However, for assay verification especially of the newer Viramed-test-platform, 192 samples collected before 2019 from patients that were presumably exposed to MERS and SARS-CoV-1 (DRK-Blood donor service Lower Saxony) and 63 samples of SARS-CoV-2-positive/negative tested persons (verified with two different systems by an official diagnostic laboratory, AG Limbach Lerthe) were used (see page 8).

December 20, 2021

Prof. Albert Osterhaus
Erasmus Medical Center
Department of Viroscience
Rotterdam
Netherlands

Re: Spectrum01512-21R1 (Combined prospective seroconversion and PCR data of selected cohorts indicate a high rate of subclinical SARS-CoV-2 infections - an open observational study in Lower Saxony, Germany)

Dear Prof. Albert Osterhaus:

Thank you for submitting your manuscript to Microbiology Spectrum. As you will see your paper is very close to acceptance. Please modify the manuscript along the lines I have recommended. As these revisions are quite minor, I expect that you should be able to turn in the revised paper in less than 30 days, if not sooner. If your manuscript was reviewed, you will find the reviewers' comments below.

When submitting the revised version of your paper, please provide (1) point-by-point responses to the issues I raised in your cover letter, and (2) a PDF file that indicates the changes from the original submission (by highlighting or underlining the changes) as file type "Marked Up Manuscript - For Review Only". Please use this link to submit your revised manuscript. Detailed instructions on submitting your revised paper are below.

Link Not Available

Sincerely,

Kileen Shier

Reviewer comments:

Preparing Revision Guidelines

- point-by-point responses to the issues I raised in your cover letter
- Upload a compare copy of the manuscript (without figures) as a "Marked-Up Manuscript" file.
- Each figure must be uploaded as a separate file, and any multipanel figures must be assembled into one file.
- Manuscript: A .DOC version of the revised manuscript
- Figures: Editable, high-resolution, individual figure files are required at revision, TIFF or EPS files are preferred

For complete guidelines on revision requirements, please see the journal Submission and Review Process requirements at <https://journals.asm.org/journal/Spectrum/submission-review-process>. **Submissions of a paper that does not conform to**

Microbiology Spectrum guidelines will delay acceptance of your manuscript. "

Please return the manuscript within 60 days; if you cannot complete the modification within this time period, please contact me. If you do not wish to modify the manuscript and prefer to submit it to another journal, please notify me of your decision immediately so that the manuscript may be formally withdrawn from consideration by Microbiology Spectrum.

Microbiology Spectrum

Paper Submission Jonczyk et al.

“Combined prospective seroconversion and PCR data of selected cohorts indicate a high rate of subclinical SARS-CoV-2 infections”

Point-to-point Response to Reviewers

Comments from reviewers are formatted bold, our responses are colored blue. All page numbers stated within this document refer to the “Marked Up Manuscript – For Review Only” document provided alongside this response to reviewers.

Line 67: Antigen or antibody?

We speak of rapid tests for screening acute infections with SARS-CoV-2, therefore we substituted the word *antigen* now by *rapid*: rapid tests.

Line 290: Do you mean December 2020 instead of December 2021?

Of course, this was in December 2020.

Thanks, best Christmas wishes! C. Blume

Prof. Dr. med. Cornelia Blume (apl)
Institute of Technical Chemistry
Callinstr. 5
D-30167 Hannover
hannover.de

E-mail: blume@iftc.uni-hannover.de
Phone: +49 (0)511 762-2963
Fax: +49 (0)511 762-3004
Website: <https://www.tci.uni->

January 7, 2022

Prof. Cornelia Anneliese Blume
Leibniz University Hannover
Institute for Technical Chemistry
Callinstrasse 3-5
Hannover, Lower Saxony 30167
Germany

Re: Spectrum01512-21R2 (Combined prospective seroconversion and PCR data of selected cohorts indicate a high rate of subclinical SARS-CoV-2 infections - an open observational study in Lower Saxony, Germany)

Dear Prof. Cornelia Anneliese Blume:

Your manuscript has been accepted, and I am forwarding it to the ASM Journals Department for publication. You will be notified when your proofs are ready to be viewed.

Sincerely,

Kileen Shier
Editor, Microbiology Spectrum
